# Perceptions and experiences of daily and long-acting pre-exposure prophylaxis (PrEP) among MSM in India

**Harsh Agarwal** [1]*, **Ivania Núñez** [1], **Lauren M. Hill** [1], **Karine Dubé** [2], **Abigail Knoble** [1], **Oluwamuyiwa Pamilerin** [1]

**1** Gillings School of Global Public Health, University of North Carolina at Chapel Hill, Chapel Hill, North Carolina, United States of America, **2** Division of Infectious Diseases and Global Public Health, School of Medicine, University of California, San Diego (UCSD), San Diego, California, United States of America

* harshagl@ad.unc.edu

**Data Availability Statement:** All relevant data are within the paper. The study's final de-identified code report extracted from software is publicly

## Abstract

Oral pre-exposure prophylaxis (PrEP) is an effective HIV prevention strategy with high efficacy. An increased willingness to use PrEP among at-risk Indian men who have sex with men (MSM) population has been reported; however, little is known about their real experiences that guide their key motivators, facilitators, and barriers with using oral and Long-Acting (LA) Injectable PrEP. We recruited participants (n = 18) through active MSM social media networks. The interviews were conducted using teleconferencing software. Interview topics included participants' sources of knowledge, experiences accessing daily PrEP, safe-sex behaviors while on PrEP, barriers and facilitators around PrEP, long-term goals with PrEP, perceptions around LA PrEP, and discussion around decision-making to switch from oral PrEP to LA PrEP. Transcripts were coded according to interview topics, and key themes were analyzed using a topical data analysis approach. Participants (median age 28 years, ranging from 18 to 40) reported a high perceived risk of HIV. This increased perceived risk was a key motivator for oral PrEP uptake. Most participants shared that they accessed PrEP through non-profit organizations serving the MSM community, where doctor consultations and medication were either free or subsidized. Even when participants reported high adherence to oral PrEP, they expressed discontent with the requirement to take it daily. High condom usage was reported concomitantly with PrEP use among most participants, except for a few who preferred bareback sex. Participants shared that they would prefer to continue PrEP until they are in a monogamous, exclusive relationship. The cost and affordability remained recurring themes, and while participants could afford oral PrEP, they wished it was cheaper, making it more accessible to the community at large. Most participants did not have prior knowledge of LA injectable PrEP but welcomed the intervention, alluding to cost and availability as the key decision-making factors affecting switching. MSM from India we interviewed adopted oral PrEP as an additional HIV prevention strategy to condoms through non-profit and private channels. Cost and broader access remains a concern among the MSM community. More acceptability research about long-acting PrEP is needed among MSM in the Indian context, and it is recommended for government interventions to include oral PrEP groups at higher risk of HIV acquisition.

**Funding:** The authors received no specific funding
for this work.

**Competing interests:** The authors have declared
that no competing interests exist.

# Introduction

National AIDS Control Organization (NACO) reported the HIV prevalence as 0.21% in 2021 among all adult population (Age 15–49) in India [1]. Separately, the 2021 HIV sentinel surveillance data reported a similar HIV prevalence of .2% [2]. At the same time, the sentinel survey also reported HIV prevalence among men having sex with men (MSM) as 3.3% [2]. Several other studies in the literature also observe that the MSM group has a higher prevalence and burden of HIV in India [3–5].

Pre-exposure prophylaxis (PrEP) is an effective HIV prevention method in addition to condom use among populations at higher risk of HIV acquisition like MSM or men in same-gender relationships [6]. A systematic review that included 15 randomized control trials found a risk ratio of 0.25 for the MSM population and also found that PrEP efficacy was significantly associated with adherence [7].

In India, PrEP is not yet a part of the prevention strategy of NACO and it is available through private doctors or non-profit organizations serving the MSM communities. India currently only has oral PrEP (Truvada (TDF/FTC)) approved and injectable cabotegravir has not yet been approved as a PrEP method. In 2022, there were a total of 2104 people who had initiated oral PrEP in India [8].

High eligibility (92.9%) and willingness to use oral PrEP– 72.9% (n = 197)–have been reported in India among the MSM population [9]. Despite this, little is known about the perceptions and experiences of Indian MSM community members who are the early adopters of PrEP. Similarly, researchers have documented high acceptability for long-acting (LA) injectable PrEP among MSM in high-income settings [10]. Yet again, there are minimal insights on the acceptability of long-acting PrEP in India.

To address these gaps, we conducted a qualitative study to explore the motivators, facilitators, and barriers around oral PrEP for MSM in India. Our study also uncovers their needs to continue PrEP, should they adopt oral PrEP as a long-term HIV prevention strategy, and examines the acceptability and factors influencing the possible eventual uptake of long-acting injectable PrEP in India.

# Methods

## Participant recruitment

We used purposeful sampling to recruit MSM participants who were on PrEP or had used PrEP in the past through MSM-specific channels on online social media platforms including Grindr, WhatsApp, Facebook, and Instagram. IRB-approved recruitment flyers and text messages were circulated through these channels, and interested participants were screened for eligibility. We included participants who identified as male and self-reported that they sexually engaged with other male partners, were 18 years of age or older, had access to smartphones or computers with the internet, and were either on PrEP at the time of the study or had used PrEP in the past.

All interviews were conducted through Health Insurance Portability and Accountability Act (HIPAA) compliant teleconferencing (Zoom). Before the start of each interview, the interviewer read and discussed the IRB-approved informed consent form section by section and answered the questions participants had. Participants were also given more time if required to make the decision. Once the participants decided to take part in the study, they provided verbal consent to participate in the study and to audio record the interview. The interview started after the consent was given to participate in the study and to record the interview.

## Data collection

Interviews were conducted between February 2022 to July 2022, ranging between 30–40 minutes. Interviews were conducted using a semi-structured interview guide. Topics included in the interview guide were based on a literature review encompassing local and global studies around MSM and daily or LA PrEP use or perceptions. The literature from India informed us about hypothetical acceptability, motivators, barriers, facilitators, and key considerations that strongly informed the key topics we included in the guide.

We sequenced these topics in a participant journey format, asking questions in the same step-by-step process they accessed daily PrEP: our interview guide first explored the participant's knowledge of PrEP and initial impressions of PrEP followed by an exploration of their key motivators to initiate PrEP.

After that, the interview guide covered how they found the provider who prescribed PrEP to them, their experiences with the providers, what could have gone differently, and their perception of the overall process. We also included questions about in-person and online provider visits since many participants started PrEP during COVID-19 and were compelled to do only online visits. We nested questions about barriers and concerns within these themes to learn what kind of barriers or concerns participants faced and how they overcame them. Other themes included perceptions around costs, sexual behavior while on PrEP, adherence and long-term goals with PrEP and LA PrEP.

## Data analysis

All interview recordings in English were transcribed verbatim by professional transcribers. Hindi interview recordings were transcribed in Hindi and then translated into English for analysis. Transcripts were de-identified, and the study team did a quality check to ensure accuracy, completeness, and fidelity in the final transcripts. We analyzed the data following a thematic analysis approach. Two investigators independently coded each transcriptin Atlas.ti using a codebook, reflecting interview topics and the recurring themes observed in the data. The two coders piloted the codebook to check the codebook application, the results were compared, and required adjustments were made to codebook definitions to ensure coding consistency. Two coders (HA and IN) then independently coded the transcripts in Atlas.ti and resolved coding discrepancies by consensus.

We then exported the code reports and analyzed the data to identify themes in participant responses within each coding topic. We then identified overlapping and intersecting themes and collapsed them into one, resulting in 9 broad themes. We then summarized the findings using narrative descriptions based on reading.

## Ethics statement

The study was approved by UNC-Chapel Hill (IRB Number 22–0053) and a SIGMA IRB (IRB Number 10086/IRB/21-22) in India.

## Results

### Participant demographics

Of 18 participants, 14 preferred English as the language for the interview. Most of the participants were employed at the time of the interview (n = 15), identified as gay (n = 13), and had completed a graduate or a postgraduate course (n = 17) at the time of the interview. Participants mostly resided in urban metropolitan areas of India, belonged to the general caste, and had an income.

**Table 1. Demographic characteristics of Indian MSM who used PrEP.**

| | Participants n = 18 |
|---|---|
| Age in years, Median (range) | 28 (18–40) |
| Duration in months on PrEP, Median (range) | 10.5 (1-48) |
| Education | |
| Graduate | 9 |
| Postgraduate | 8 |
| Intermediate | 1 |
| Income in Indian National Rupees (INR) | |
| 0 (Unemployed) | 2 |
| Up to 30,000 | 4 |
| 30,000 to 60,0000 | 2 |
| 60,000 to 1,00,000 | 6 |
| Above 1,00,000 | 4 |
| Caste, n (%) | |
| General Caste | 14 |
| Other Backward Classes (OBC) | 3 |
| Scheduled Class (SC) | 1 |
| Sexual Orientation | |
| Gay | 13 |
| Bisexual | 3 |
| Queer | 2 |
| Sexual Preferences | |
| Top (Prefer insertive anal sex ) | 5 |
| Bottom (Prefer receptive anal sex) | 4 |
| Versatile (Prefer both) | 9 |

The median age reported was 28 years. None of the participants reported being completely closeted and were out to their friends, close family members, or everyone. 16 participants reported that they were single, 1 in a relationship, and 1 shared that their relationship status was *Its complicated*. We describe participants demographics in Table 1.

13 participants were using PrEP at the time of the interview, 4 participants had used PrEP in the past and were no longer using it, and 1 participant reported using PrEP on Demand model. The median duration on PrEP was 10.5 months (minimum being 1 month and maximum being 48 months). We describe our key findings in the below section. Further we summarize the key themes and illustrative quotes in Table 2.

## Key motivators for using PrEP

Participants shared a variety of motivators for PrEP initiation and persistence. For example, many participants discussed the perceived high risk of HIV, not knowing their sex partners' HIV status, and having peace of mind to be the key reasons they had decided to initiate PrEP.

> "*What if he is positive, you don't know, people don't discuss these things when they are meeting a person. They just want to hook up and then suddenly this came up in my mind that, okay, why not why don't I try PrEP, though I use a condom.*"—P09, Age 28, Gay.

**Table 2. Key themes and illustrative quotes.**

| Theme | Illustrative Quotes |
|---|---|
| **Theme 1: Key Motivators** | *"There are a lot of guys who don't want to use condom, and so for those times i'm really glad that. You know this is something i've been on."*<br>*"A lot of positive homosexuals moving around having zero safeguard issues for themselves and for others right, so I wanted to safeguard myself as priority, just in case there is the issue or a slip that happens right."*<br>*"PrEP does give me the blanket. the umbrella of safety."*<br>*"Last year, my VDRL was positive so I had to take an injection. So, that's why I thought it will work for me because my sex encounter was on a large scale. So, to prevent myself from all this, I thought prep is a good option."*<br>*"I met a guy who is HIV positive in his initial stages. I was..I knew, I know that guy since 5 years. We are very good friends and initially I was not that sexually active and we both became sexually active in 2020 after the first lockdown"* |
| **Theme 2: Concerns and dislikes around PrEP** | *"The KFT test, there's a Kidney Function Test. I, yes, I did have an increase on few pointers. So I will be doing my..and that's one of the reasons I stopped this start of the year, because I wanted to see whether, if I stopped prep, will it come down, and it did."*<br>*"For me time is an issue like it's to be taken at the same time everyday. Like, I started to take it at night and my doctor asked me to take it at that time everyday. So, I just wish there weren't any time boundations and that we can take it at any time of the day"* |
| **Theme 3: Experiences during accessing PrEP** | *Prep was something which was introduced to me by a friend. He is from the US. And he was visiting India*<br>*And then I was traveling to US in 2017–18. That's when I met a lot of people who were on prep*<br>*I also got an ad on my.. on my instagram I think. Either instagram or Facebook regarding this organization that helps provide prep to.. to queer people in India*<br>*I started getting more information about some of the NGOs having the alternative of partnering with the companies here to get prep.*<br>*"He (doctor) actually spoke to me for half an hour. He explained everything and then it was quite helpful and he sort of broaden my perspective, about what's the pros, cons, everything is and that's when I was a little more comfortable about getting on with it."*<br>*"Preferably, in person. . .I can meet online as well, but it's.. it just helps to first meet in person, and then you can follow up online."*<br>*"It wasn't available at any of the medical stores near my house. I tried one or two, but they had no idea what prep is. Maybe if I had said truvada then they must have understood but not when I said prep."* |
| **Theme 4: Risky Behavior with PrEP use** | *"No, definitely. I'll use condoms when I'm penetrating anyone, definitely I will. I need to use, there is no exception over there."*<br>*"So, initially, I always used to use protection even with prep, but I think recently, probably from the past month, I have become a little more relaxed about it"*<br>*"Honestly I do not do anal intercourse with everyone. So, uh..and when I do it, it's like 50–50 depends upon the person, right, as I said, so sometimes I feel like not using condoms and sometimes I feel like I should use condoms."*<br>*"Mostly, I did not. Only sometimes. It's like if the other person gave it to me or asked me to use it, only then because I don't like using condoms as such."*<br>*"Like before I didn't know about the STIs, I was like I won't use condoms if I'm going to get on prep, but then the STIs thing came.. I was like.. okay, I have to use condoms. Okay, yeah and condoms are kind of fun."* |
| **Theme 5: Adherence and Trust with PrEP** | *"I just read it online and I've gone through the 2-1-1 method. I read something about prep online and well, it was promising. So yeah. I wanted to get on it. People, I've seen people who are on it. I've heard stories from them, what they did. Without a condom obviously. And I felt like, they're safe so, yeah."* |
| **Theme 6: Long Term goals with PrEP** | *"I think the reason would be the same if I'm dating somebody and we decided to be monogamous."* |

(*Continued*)

**Table 2.** (Continued)

| Theme | Illustrative Quotes |
|---|---|
| **Theme 7: Cost and affordability** | *"I mean anything cheaper.. being cheaper is great, I feel. I can afford it, as of now, I don't mind that cost. I think it's a great price cost,. . .but I'm sure there are other people who can't afford it"*<br>*"Of course, it's not an affordable drug per se, to the Indian market."*<br>*"I do wish it was cheaper, but I like, relatively it is affordable. But I think for more people to adopt it, it needs to be cheaper. It's too expensive."* |
| **Theme 8: Community and peers' reactions to PrEP use** | *"Being my closest friend he reacts in that way, like what happened to you why are you dating that guy? Why are you troubling your body?"* |
| **Theme 9: Long Acting Injectable** | *It's a very good idea, because having a pill, you might forget your pill any day like it happened to me once, yeah I missed and it's also a hassle like every day, taking a pill, if you're outside your house, you have to always carry your pill, and going somewhere else.*<br>*I really want it to come very, very fast because taking tablets every day is a.. I don't know, I just don't like it.*<br>*Yeah, I would like to know more about its side effects, adverse effects and all that.* |

The participant alluded to the lack of discussion around safe sex, STIs, and HIV in the community and how it is common for people to sexually engage without having these conversations. Another participant talked about the lack of testing among the MSM population.

"*And I was hearing rumors that there are a lot of boys people around here, who are not informative about their status, I would say and but even though I am being and using condoms, and everything but it's always better to be one step further and safe.*" P05, Age 26, Gay

The sentiment of remaining safe with PrEP as an added prevention tool resonated with many participants. One participant also stated that they did not want to be biased or discriminate against those who are living with HIV for sexual encounters, which was one of their reasons for taking PrEP.

"*So regardless of the person that I have met was positive or not, I would still end up having sex with them, is what I mean like I would still be open to having sex with them or meeting them or you know, taking things forward, but their sexual, their health status wasn't a prerequisite for me deciding because that's basically discrimination again.*" P02, Age 28, Gay

Few participants also shared having a partner with HIV with whom they had or wanted to build a romantic relationship.

"*He's HIV positive but I was attracted towards him and. . . that was the. . . like you can say start, I had to do PrEP so that obviously for today I am doing sexual things with him, I am getting sexually attracted so uhh. . . I should be safe on my side. So. . . I started PrEP because of him.*" P03, 25, Gay.

The desire to stay in a relationship with an HIV-positive person became a key motivation here. This participant was able to overcome the stigma around the fear of being with an HIV positive person and take steps to protect themselves as well. Uniquely, a bisexual participant also highlighted social concerns about being HIV-positive and hence they wanted to use PrEP to protect themselves.

"*Not about scared of. . . about being positive or all because we know we have very good medi-cations now. But the thing is the social concerns. The marriage and all these things.*" P011, 28, Bisexual.

The motivation to remain negative here was not the fear of HIV but the understanding that their desirability as a potential suitor in an arranged marriage setup would decline, should they become HIV positive.

There were also a few participants who were clear that they wanted to use PrEP since they preferred bareback sex and did not want to use condoms.

"*It's a good thing in a sense. . . That I can also bareback*" P18, 26, Bisexual.

"*Other person gave it to me or asked me to use it, only then because I don't like using condoms as such. This is my main reason to use PrEP.*" P08, 27, Gay.

Other participants, while stating ambivalence about condom use with PrEP, didn't explic-itly express a desire for bareback sex as a reason to use PrEP.

### Experiences accessing PrEP

Participants' experiences differed as per the route through which they accessed PrEP. How-ever, the most common route to seek PrEP was through NGOs that offered PrEP at subsidized or no cost.

Source of information: Participants identified multiple sources from which they first heard about oral PrEP. Many participants reported knowing about PrEP when they were visiting a Western country or from people visiting from those countries. Others reported finding about PrEP from their peers in India, through dating applications, and non-profit organizations serving the MSM community.

"*So when I went to the UK, which has been in 2018 to do my masters, that's when I really heard about PrEP in the local sexual health clinic.*" P02, Age 28, Gay

"*My friend, he told me about this foundation. . . [Name of NGO]. They were giving PrEP for the. . . for students for free so that's how I came to know.*" P17, Age 18, Queer

"*See if I talk about online, I am on Grindr. I keep seeing profiles, it says negative and on PrEP. Okay, so we obviously, people are curious, so I started reading about it online.*" P09, Age 28, Gay

This illustrates how, in the absence of reliable sources of information that are known and accessible to the community—friends, peers and the online universe becomes a key influencer in gaining knowledge and finding access to HIV testing and prevention services.

Few participants felt that it was hard to find PrEP in India when they started searching and that PrEP was expensive, which appeared to be one of the barriers to initiating or staying on PrEP.

"*I did try, it wasn't easy, and the availability was almost zero. Where I did research where I can get it, which doctors can give me a prescription, and how I can go about it. Zero informa-tion around it.*" P12, Age 36, Gay

Trust with PrEP: All participants said that they trusted PrEP to protect them from acquiring HIV. Participants cited a variety of reasons for why they trusted PrEP. Reasons for trusting

PrEP included reading online, hearing from peers that it works, noticing more conversations about it in India and globally, and because of the counseling they received from PrEP-related visits.

"*I did [trust PrEP], because I think there's enough research that suggests that it works. So I'm not like a conspiracy theorist.*" P14, Age 40, Gay

Participants trusted the science and research behind PrEP which increased the trust. Other participants researched online to develop trust.

"*Because I Googled and there I got to know that it's quite trustworthy because it's human nature, you can't just trust words. So, when my doctor told me even at that time I wasn't confident enough but then I Googled it for nights. I did not take my medicine for 2–3 days when I bought it because I wasn't sure. But ultimately I had to because my friend was taking it and I had good reviews too.*" P08, Age 27, Gay

However, simply looking it up online didn't suffice for the participant above. They didn't trust PrEP even after finding access to it. For them, the critical point was hearing good feedback from their peers–connecting back to the role of peers in influencing opinions and decisions about PrEP use. Another participant shared similar comments and that their confidence increased because the people they knew informed them that PrEP was working.

"*There's a lot of conversation that's going around it. Extensively in global discussions. Like a lot of social media influencers are talking about it as well. So, that element of confidence that they are talking about it, that it's working for them, their friends, the people that they have around.*" P12, Age 36, Gay

One participant highlighted the vital role that providers can play in further strengthening trust and confidence in PrEP.

"*I think the sessions we had with the doctor and like he explained how it attacks the virus if we get. . . like.. in contact with the virus. So, I think it was pretty believable.*" P17, Age 18, Queer

Experience with providers for prescription: Most of the participants reported getting connected to a PrEP provider and eventually obtaining PrEP through various non-profit organizations in India working for the Lesbian, Gay, Bisexual, Transgender, and Queer + (LGBTQ+) community, with few exceptions of those who obtained PrEP through private practices (offline and through telemediation). Participants mentioned organizations like 1mg (online pharmacy and telehealth) and Practo (booking offline doctor appointments and telehealth) to access providers. Participants reported high satisfaction with the process and found it simple.

"*It was [Name of NGO], whose ad I had seen, and I. . . I had somebody's number from [Name of NGO], so I just reached out to them and told them that you know I had, I had an HIV scare, and I want to get on PrEP and they were very helpful, we went ahead and I got tested and everything and all that and that's how I got on PrEP.*" P07, Age 23, Gay

"*So, I called the NGO and the process was pretty simple. I had to give my blood sample for my kidney test and HIV test. The whole process was, like, quite simple and when I started,*

*although I was a bit hesitant about it but then I was like, the whole process turned out to be quite smooth. I didn't even have any symptoms as such.*" P04, Age 28, Gay

The participants felt comfortable with the NGOs and again, their own peer network played a critical role in making those connections with the NGOs. Indian online pharmacies are essential players in the Indian healthcare market and our study participants also reported trying to use those avenues to access PrEP.

"*Do you know 1mg app in India? So I actually got my test done and then someone on that app recommended like, gave me a prescription for PrEP.*" P09, Age 28, Gay

One participant also asked their regular provider for PrEP,

"*I used to go to him (private doctor) for normal checkups as well. So, he said it cannot be taken just like that because he did not know about my sexual intercourse because sex is a topic people don't usually talk about. And I was scared to ask as well because of the fear of being judged. Still, I asked him and he asked me to get tested and told me that HIV needs to be negative before starting the medication. And all my tests were, so he said I can take it.*" P08, Age 27, Gay

However, the lack of confidence in discussing PrEP with their regular provider gives a glimpse of the perceived stigma and judgment that they may face from the provider.

Many participants appreciated the providers they met who prescribed PrEP and mostly reported a positive experience. Participants met their providers in both online and offline settings. Discussions indicated that the overall preference for meeting providers was in person for future appointments.

"*I wouldn't say there was anything that I didn't like about it. He was very good, he was supportive, he made me comfortable, comfortable enough to talk about my sexual life, my partners, my history, etcetera and it was pretty chill.*" P07, Age 23, Gay

Feeling comfortable appeared to be an important parameter for the participant–which connects back to the perceived stigma and judgment from the providers that few other participants mentioned.

"*I would prefer in-person. . . I think it's more comfortable talking to him in person than on call. Because it will be in a closed room and not on the phone, where someone can hear you.*" P09, Age 28, Gay

The participant was worried about their privacy being violated if they were engaging with the provider online–putting them at risk of accidentally coming out to their cohabitants. One participant felt that their provider was robotic, and that the provider was in a rush. They elaborated,

"*I would say the explanation was very robotic. It was more of like, this. . . this. . . this. . . this. . . this. . . this. . . and go thing, sometimes it was more of like and I was also working at the same time, I asked him if I can speak later, but he was like okay, we have to reschedule and everything. So what happened is it was very, very robotic, so I am not sure if it's the same for everyone, but I felt like it would be like, if he can take some time and have a proper consultation. It would have been much better.*" P05, Age 27, Gay

Participants clearly demonstrated a need for a good experience with their providers where they felt comfortable, their privacy was protected, and they were given sufficient time to review the details.

Experience accessing the pills: To obtain PrEP, once participants had a prescription, participants mostly continued to rely on using non-profit sources for subsidized rates. Those who used non-profit sources reported PrEP to be home-delivered. Some also directly purchased PrEP from eCommerce websites, local medical stores, or the providers they received the prescription from.

"*I got some understanding from him (friend) that the NGO that he was working with... was into getting people tested, they have some things around... they can arrange for me to have a discussion with the doctor and post which, if everything is good, I can get on to PrEP by making a certain amount of payment. Subsidy amount that is there for PrEP in India, right. So what I do is I make a payment to them (to NGOs) and give them my address, and the details that they need and then they send it over.*" P12, Age 36, Gay

Participants appeared to enjoy the convenience of home delivery, making it a preferred option; and also appreciated the option to pick it up from the pharmacy.

"*Oh, it's Medplus... it's Medplus pharmacy. We have an app for it. So, they do have all the medications and everything. It's like I could just go there and get it by myself, or I could just place an order online and they could deliver it to my house or I could just pick it up from there.*" P15, Age 31, Gay

## Concerns and dislikes around PrEP

The participants reported no significant concerns or dislikes that could lead them to be swayed away from using PrEP. Some participants discussed their concerns about the potential side effects of PrEP. However, very few reported any side effects once they initiated oral PrEP.

"*I think the initial concerns that were highlighted online and by others, that you will get nauseous, you might have a headache, you well... you might have stomach ache, cramps, stomach upset.*" P12, Age 36, Gay.

While sharing above, the participant also indicates that online mediums and other peers from the community were their sources of information and can influence their opinion about PrEP. Another participant was relatively more concerned and had similar knowledge about potential side effects.

"*I was concerned if it would have any side effects, you know because I have heard it has side effects like it might affect your kidney in the long run, might affect your liver, in the long run, so it was a little concerning, for me, and on top of that, I do not have intercourses as often or I'm very, very picky and I'm very choosy about my partners in that case.*" P05, Age 27, Gay

An additional concern about not engaging in sex frequently also emerges above. Participants expressed further concerns about the commitment of PrEP as a daily oral regimen and were unsure if they could keep up with the commitment.

"*It's quite a commitment, and you can't forget about it, and you know you still have to be careful and... plus I read some things about how like, in the longer run it may affect your*

*liver function, your kidney function and so you have to like constantly take care of that. I feel that I have not just not sleeping with anybody for like say a month or two months, and I am still on my PrEP and I'm just taking it, taking it, taking it. So these other things that you know, at a point I just start wondering why am I even doing this, like what is the point.*" P07, Age 23, Gay

This further illustrates the concerns about taking PrEP daily for it to be effective even if they are not engaging in any sex activity frequently or for a long time. Participants also talked about being more lenient with safe sex or behavioral disinhibition which they felt was one of the concerns with PrEP use.

"*The downside to that is also when you're on PrEP and if you're drunk or you are high or anything, then you tend to be. . . then you. . .then you. . . if somebody is on PrEP too, then you tend to be a little more loose about rules. About having safe sex and then you're like okay if you're on PrEP and the other person says that he's on PrEP, then you do slip one once or twice*" P13, Age 31, Gay

The above indicates that participant's propensity to not use condoms increases when under the influence of alcohol or other substances, and if they are on PrEP. It is worth noting that the participant did not consider using PrEP alone as a safe sex practice. Overall, in our study, participants reported minor concerns and dislikes–none of which deterred them from continuing to use PrEP. The increased propensity to not use condoms also didn't result in the discontinuation of PrEP.

## PrEP use and dual protection

Most participants reported preferences to use condoms while using PrEP with some exceptions who did not prefer to use condoms and hence had started PrEP. Among those who preferred condoms, there was a certain ambivalence with the partners they knew or were not one-time or with random sexual encounters. Participants shared that they were comfortable to not use condoms if they knew the person or had met them a few times.

"*Well, I mean it all depended on the partner if they were trusted partners and friends and friends of benefits that I knew them then it would of course be without a condom. But if it was just any other random hookup it would have used protection.*" P02, 28, Gay

Participants felt confident about not using condoms with people they knew. However, there was no clear understanding of how those were defined. Using PrEP perhaps gave them the confidence to not use condoms with *known* people in their network and they were willing to take a calculated risk for potential STIs–the selected condom use while on PrEP was reported by other participants as well.

"*I always end up using condoms with like new people or hookups or like even if it's a repeat hookup, I usually don't. . . but the two partners that I told you about who is fairly like, regular initially we used to use condoms, but then slowly became so that you know, we got comfortable with each other, and since I'm on PrEP, so the beliefs, with these two people I stopped using condom.*" P07, Age 23, Gay

Another participant explained that during most of his sexual encounters, they didn't use condoms and only used them if their partner insisted.

"*So, if I had 10 encounters, then 7–8 would have been bareback, and in few, it was like there is no option. When your partner is concerned that it's okay you are on PrEP but we will still use it, then in that case I used to use it.*" P18, Age 26, Bisexual

Most participants had knowledge about the risk of contracting STIs in the absence of condoms.

"*Like before I didn't know about the STIs, I was like I won't use condoms if I'm going to get on PrEP, but then the STIs thing came. . . I was like. . . okay, I have to use condoms. Okay, yeah and condoms are kind of fun.*" P17, Age 18, Queer

This young participant shared that they didn't know about STIs at all, depicting the need for more awareness around STIs and including messaging about STIs in existing HIV prevention programs and messaging. Another participant was upfront about the risk of STIs and had clarity that PrEP didn't protect them from those STIs.

"*Like, PrEP is going to save me from HIV but there are other diseases like syphilis and chlamydia and everything.*" P04, Age 28, Gay

Few participants shared that they indulged in chemsex (use of chemical substances to enhance sexual pleasure) which led to distorted judgment about condom use and safe sex.

"*I was also in the drug scene for a long time and in that time you know a lot of the tops prefer. . . either prefer not to wear it or by the time they're high they're not hard yeah so as a result of that it's very difficult to convince them to get on to use a condom. . . so a lot of them were not wearing condom and it was. . . it was difficult to have that negotiation when you're in that space, yeah so I think yeah so I. . . if it was me, I would have preferred it, but it almost. . . it almost never happened.*" P16, Age 40, Gay

This alludes to how when using aphrodisiacs, it is hard to have conversations and negotiations about condom use, putting the participants at risk. Another participant resonated with this and expressed similar concerns.

"*When you're high it's very unlikely that you're going to remember, or even choose to remember that you know. Precautions and contraceptives are necessary, and you know all of these things like protection is necessary But, eventually, you know, like an addictive mind kind of go haywire and you still don't use it.*" P02, 28, Gay

Interestingly participants reported that lack of condom use didn't increase with PrEP use. They saw PrEP as an additional prevention tool and not the only one.

## Adherence

Most participants reported high adherence to daily oral PrEP. Some shared that they missed a few doses but none of the participants reported frequently missing doses. Some of the reasons shared by participants for missing doses included being drunk, coming late to home late, traveling, delayed delivery by the provider, or simply forgetting about it sometimes.

"*I come back, maybe wasted drunk or something, yeah then that, that night, once it has happened, that I forgotten to take the night dose, the night when I've seen it in the morning, so*

*I've taken it immediately in the morning but that thing still kind of broadly counts."* P16, Age 40, Gay

*"Never.. like maybe taking a dose later when I might have been out, but then that's like maybe two hours later, something like that, but that's about it or unless where, there was a delay from the logistic partner and, yeah."* P12, Age 36, Gay

While participants showed high adherence, they remained susceptible to missing doses occasionally. Participants shared that they use tools like pill boxes, alarm clocks, keeping PrEP with other medications or supplements, or help from friends to get reminders. Some shared that they have integrated PrEP into their habit, facilitating \ adherence.

*"I usually don't (forget) unless something is going on. I'm busy at work, I have an alarm on my phone that I keep every night. So I know that I have to take it if I'm going out, probably on a day before I have to go out."* P14, Age 40, Gay

*"For me, based on my habits and my lifestyle, I think I am that kind of person who. . . who adheres like, who has this habit of being compliant. It's become a habit, I take it just before I brush my teeth before going to bed."* P15, Age 31, Gay

Overall, Participants reported high self-regulated adherence, indicating that they grasped the responsibility that comes with PrEP.

## Long-term goals with PrEP

Many participants reported that they had discontinued PrEP in the past. The two significant reasons shared by participants for discontinuing PrEP were sexual inactivity because of the COVID-19 pandemic that posed requirements for physical distancing and getting into a relationship. A few participants also shared that they discontinued since they moved back to their hometowns from a bigger city that would limit their sexual activity.

*"I started; I would say in. . . I think I started in 2018. Then, I was on it for about a couple of years until like COVID started getting serious. So maybe until like May of 2020 when it was clear, like there was no sexual life happening. And then I restarted it last year."* P16, Age 40, Gay

*"When I went to my hometown from Delhi, after that when I came here, like 'til I visited my hometown. . . Actually, that time I was really like into sexual intimacy. After that, when there was nothing."* P06, Age 25, It's complicated

The PrEP use for these participants revolved around their sexual activity. They initiated and stopped PrEP use if they were not sexually active–this could be because of affordability, having to remember to take a pill, or simply the worry of using a medicine that they don't need as such. A participant mentioned that they stopped using PrEP when they were in an exclusive relationship.

*"We decided to make it kind of official and exclusive for a bit and so that happened, but then didn't work out and we broke up soon after. So then in between the period when, where we decided we will get exclusive to the breakup. That is when I decided that I will not take PrEP anymore, because there's no point."* P07, Age 23, Gay

The role of finding a partner and getting a relationship is linked to PrEP discontinuation. We further see that other participants also share that they would prefer to continue using PrEP till they find a partner. At the same time, one participant discontinued because of concerns about elevated results in the kidney function test (KFT).

"*I just decided that hey you know let's just try, considering that my KFT was showing high. Let me just do one thing, let me just take a breather and then see how it impacts me and my health.*" P12, Age 36, Gay

Despite these breaks, most of the participants who discontinued eventually re-initiated PrEP. When probed about what their long-term goals with PrEP would be, most of the participants shared that they would prefer to stop PrEP if they got into an exclusive monogamous relationship and had a partner. Some participants said they would continue PrEP until a vaccine for HIV is found. Many shared that they would take PrEP to the point where they could afford it and their body could tolerate it.

"*So either. . . if I get into a relationship which is exclusive or. . . or if a vaccine comes out, otherwise I plan on staying on PrEP for a bit.*" P07, Age 23, Gay

"*If I can afford it in the future as well, then I will continue for a lifetime and then secondly, it doesn't affect my body.*" P08, Age 27, Gay

The theme of affordability was observed across questions and appeared to be linked with the decision-making process around PrEP. We explore it further in upcoming sections. One participant said they would prefer not to discontinue PrEP and will continue to take it for a lifetime if possible.

"*I don't plan to get off it even if I'm less sexually active than before, or even if I do have like a committed partner, you know, whatever. I've actually never been monogamous, so. . . yeah, I think, even if I were to have like a steady partner I wouldn't get off it so for now yeah, I imagine, I would take it for the rest of my life.*" P16, Age 40, Gay

Participants overall appeared to remain committed to PrEP to protect themselves from acquiring HIV.

## Cost and affordability

Participants reported a range of costs for PrEP and provider consultations. Those accessing PrEP through NGOs shared that their provider consultations were mostly free. Some NGOs provided PrEP for free while others provided it at a subsidized rate that ranged between INR 700 (USD 8.75) to INR 900 (USD 11.25) for one month of PrEP. Those who purchased directly from medical stores (online or offline) shared a wide range of INR 1200 (USD 15) to INR 2500 (USD 31.25) for one month of PrEP.

Most of the participants felt that PrEP should be cheaper. Even if they could afford it for themselves and found value per the cost they were paying, it remained inaccessible for many others from the community because of the high cost.

"*It's not affordable at all, and I think. . . I've been given. . . my class circumstances I still thought this was a huge sort of. . . not an impact on my wallet but it was certainly significant*

*like it's not a small amount of money, yeah, so I do, yeah I do think it's not affordable.*" P16, Age 40, Gay

Participant's perception of affordability went beyond their own financial standing, addressing the economic diversity in the Indian MSM community and the fact that currently, PrEP is paid all out of pocket, which means that only people of certain financial standing will be able to access it. Another participant similarly shared,

*So I wouldn't say it's like super cheap also but it's affordable for me and but the concern could be that, for some people out there it could be too expensive. Even at the subsidized rate.*" P07, Age 23, Gay

The participant felt that even after the subsidies provided by NGOs for online consultations and drug costs, it was still unaffordable. One participant shared their need for PrEP to be available for free at government hospitals.

"*No, according to me, it should have been cheaper. I think I must have gotten it from the government hospital, that too free. Because many people can't afford it even if they are aware of it because for some people even 1000 rupees is too much and for some people it's nothing.*" P07, Age 23, Gay

While participants were willing to pay for PrEP, having PrEP available cheaper may lead to more initiations in the MSM community.

## Community and peers' reactions to PrEP use

Participants shared that they feared judgment from the community and peers if using PrEP since they might be perceived as promiscuous. However, very few participants disclosed any stigma received from the community. Participants discussed that correct knowledge about PrEP was scarce and they got mixed responses when they shared PrEP use with others. Many participants interacted with also confused PrEP with antiretroviral treatment (ART) used to treat HIV. Some others cautioned them because of side effects, costs, and other reasons.

"*Like I told my sister as well and she was curious about it and then she was a little apprehensive as to why I need to be on a pill.*" P13, Age 31, Gay

"*I was chatting online. . . He says, you are on PrEP and were you diagnosed with HIV or you are just taking it? I said if you're on PrEP, I think only negative like, was, whoever is negative, can only take PrEP.*" P09, Age 28, Gay

The confusion with being HIV positive for PrEP was shared by other participants as well. However, the participants in our study were not afraid of them being perceived as HIV positive and took the initiative to educate their peers,

"*Like I try my level best to make them understand that I am taking these medicines to prevent HIV not because I am suffering from it.*" P03, Age 25, Gay

Many participants shared that they encountered several situations where potential sex partners would suggest not using condoms since the participants were on PrEP (even if the potential partner was not on PrEP).

"*First is the. . . is the kind who knows what PrEP is. So, so that. . . then it's great like okay he knows PrEP. . . that's great, then there are people who haven't have a vague idea about PrEP, so that is the bunch who usually go like, Oh, then we will not use a condom. I think the most common one is, okay then we don't need to use condoms. Then there's a group of people who think I have something and hence I'm taking some kind of medication so. . . they, some people freak out also on. . . on learning that there is something called PrEP that I take. I think the most common one is, okay then we don't need to use condom.*" P07, Age 23, Gay

The participant's comment also suggests the prevalent misconceptions about PrEP in the Indian MSM community and the lack of correct knowledge about PrEP. Another participant also shared situations where their potential sex partners suggested having unprotected sex since the participant was on PrEP.

"*They don't want to take PrEP but they are happy that I am taking it and so, they don't want to use condoms. I ask them why don't you take it, and they say that they don't want to but you are safe so let's not use a condom. And I have no idea if they are or not.*" P08, Age 27, Gay

Consistent demand for unprotected sex from potential partners elicits the desire to engage in bareback sex among some community members while their own PrEP use is unclear.

## Acceptability of long-acting injectable PrEP

Most participants had not heard about long-acting injectable PrEP. Those who had heard about it had some vague idea about it. No participant reported complete correct knowledge about long-acting injectable PrEP. Once explained, participants showed high interest in and acceptability for injectable PrEP.

"*I find that easier than taking a pill every day, if I had the option between the two. Right now, I have the option to either take a pill every day or not at all, and if I had an option between taking a pill and an injection, I would take the injection.*" P14, Age 40, Gay

The interest in injectables links back to what we learned when discussing adherence–where people reported missing a dose occasionally. Another participant elaborated on how inject-ables could *fix* that problem.

"*It's better than everyday tablets because you can miss your dose sometimes. But with injec-tions you can fix like, I will get my dose at the end of the month, on a specific date. Because you miss your medicine sometimes, because of rush, or maybe you forget so if you fix a specific date for this injection then it's quite good, comparatively.*" P06, Age 25, It's complicated

The participant felt that fixing one day for an injection and showing up was relatively better than remembering about the pill daily. Access or availability and affordability were identified as key influencers for decision-making for the uptake of long-acting injectable PrEP.

"*If it's easily available, if it's affordable, um. . . if people have knowledge about it, especially doctors. Then it's beautiful.*"

It's interesting to note that the participant wanted the doctors to know about long-acting injectables, which would factor in their decision to take long-acting injectables. The desire for knowledge around long-acting could be possible because of the current lack of knowledge

about oral PrEP among Indian healthcare providers. We noted more participants talk about affordability

"*It depends on how much it costs. It definitely is a better option because, the way I feel about taking medicine every day, I think each person would take it, it just has to be budget-friendly. There are many students in our community who have sex, so if it's budget-friendly, it would be a success.*" P08, Age 27, Gay

The participant specifically referred to the needs of students and young people who are sexually active but might not have the means to access PrEP and pay out of pocket. Participants also asked questions about the effectiveness of this intervention and the kind of possible side effects that they might encounter.

"*And also I need to have statistical data in terms of like research done around it, where like thousand or people have tried it, it worked on and the efficacy of it right that's what I was looking for.*" P12, Age 36, Gay

This adds to the experience of other participants who shared that they trusted oral PrEP because of the research behind it. Similar to oral PrEP, side effects were one of the concerns when it came to long-acting injectables.

"*Side effects one big thing. Two is how effective it is. Is it equally effective or better than the current we have and third is basically how. . . How affordable will it be*?" P05, Age 27, Gay

Most of the participants said they were okay with an intra-muscular injection. One participant said that they would prefer to stay on PrEP pills unless the long-acting period was longer than two months.

"*Yeah, yeah. Personally I think taking a pill every day is easier for me as of now. But I wouldn't mind if it's like one injection per three months or for a longer time.*" P04, Age 28, Gay

Considering there is more research around products that might be further long-acting, acceptability and uptake of injectables may improve if they are more long-acting than the currently available products. Overall, we noted the high acceptability of LA PrEP with the caveats around cost, availability, and possibilities of having a more long-acting effect than the current regimen.

## Discussion

Our qualitative in-depth study reveals a strong commitment to oral PrEP (high adherence and long-term plan to use PrEP) by the Indian MSM community in cognizance of their high risk of acquiring HIV. We also found cost, availability, and access as important parameters around PrEP uptake decisions. Further, NGOs serving the MSM community are playing a major role in the current PrEP landscape of India. We also report high hypothetical acceptability of long-acting PrEP by Indian MSM. Multiple participants also mentioned using digital platforms for provider consultations, ordering medication, and diagnostic tests–indicating a wide acceptance of digital health service delivery.

Our findings around the perceived high risk of HIV as a key motivator to adopt PrEP are consistent with other studies exploring willingness to use PrEP in India. Chakrapani and

colleagues in their study (N = 197) also reported perceived HIV risk to be correlated with higher odds of Willingness to Use PrEP, along with greater perceived benefits and being eligible for PrEP [11]. Commitment to staying negative has also been identified as a key motivator by young MSM in the United States(N = 178) [12]. A study in Mumbai (N = 197) found that PrEP efficacy was correlated with the decision to use PrEP among MSM. This corroborates with our study results where we often found participants talking about doing their own research to know more about efficacy or asking questions about the effectiveness of long-acting PrEP [13].

A qualitative study among Thailand MSM brings forth similar concerns noted by our participants, including worries about side effects, HIV-related stigma, and affordability of PrEP [14]. Early MSM adopters of PrEP in southern Germany have also raised concerns about side effects [15]. An Indian study (N = 8621) highlighted that being perceived as HIV-positive for using PrEP resulted in an unwillingness to use PrEP among MSM [16]. While participants in our study shared the experiences of being perceived as HIV positive, this did not deter them from using PrEP.

Participants' prospective decision to stop using PrEP once they find a partner and enter a monogamous relationship reflects the desire and aspirations of this marginalized group in India where gay marriage is not yet identified. The Supreme Court of India decriminalized same-sex relationships in 2018, but the battle for equal rights to marry continues in India [17, 18]. These reflections also contradict the usual misconceptions and stereotypes about the gay community being promiscuous [19].

On the other hand, a bisexual participant's motivation to remain HIV negative for social desirability as an eligible match for a girl reflects upon several bisexuals and closeted gay individuals who maintain sexual relations with other male partners but prefer to marry a woman, have kids and start a family to meet the social norms and expectations [20–22].

Some of our findings contrast with those in high-income countries, particularly around STI and condom use. For example, MSM (N = 2300) who used PrEP in Berlin reported high STI incidence and low condom usage [23]. Similarly, contrasting to high self-reported PrEP adherence in our study, Latino MSM (N = 159) in Texas (United States) reported struggling with adherence with only 46% being able to take their PrEP daily [24]. A study in China reported an anticipated reduction in condom usage after PrEP use among MSM [25]. On the other hand, MSM in Vietnam (N = 1069) reported high (>90%) PrEP adherence [26].

We found that few PrEP users we interviewed engaged in Chemsex or High fun (mixing sex with aphrodisiac chemical substances). Closson and colleagues document the views about the adherence of those who engage in recreational drug use. Their findings reveal that such MSMs in Boston felt it would be hard to adhere to daily medications concurrent with their drug use. Participants in our study who also engaged in recreational drug use did not share such concerns [27].

Further, the literature suggests that MSM preferred subsidized rates and government hospitals against private hospitals. This coincides with our study findings where most participants accessed PrEP at subsidized rates or for free. We had no participant reporting taking PrEP from government hospitals since PrEP remains unavailable in government settings in India so far. However, our study also had a very limited number of participants accessing PrEP through private channels [13]. Cost and affordability play a role in other lower-middle-income and high-income countries as well [28–32].

Our sample had a high hypothetical acceptability of long-acting injectable PrEP in India. Belludi and colleagues' study revealed that Indian MSM did not have a high preference for injectable versus oral PrEP, with oral PrEP being preferred by 39.6% (N = 8621) and injectable by 41.7% [16]. On the contrary, our study indicates a higher affinity towards long-acting

injectable PrEP. Eisingerich and colleagues' findings that the route of administration was an important attribute for Indian MSM for PrEP use with a bimonthly injection in the buttocks having a higher score over a daily pill [33]. In addition, the literature suggests varying preferences when comparing daily oral PrEP with injectables in different countries. MSM in the United States also reported preferring daily pills over injectables [34]. On the other hand, a study in Toronto, Canada reported the opposite [35]. In Vietnam and France, long-acting injectables were also preferred over daily regimens [36, 37].

Only one participant in our study reported using the PrEP on-demand model. Another participant, when presented PrEP on demand as an option to his provider, was rejected and asked to go on a daily regimen. Considering intermittent PrEP has also proven to be effective in preventing seroconversion among MSM, it is imperative that more awareness be created among provider and patients for PrEP about the on-demand model—especially considering that the MSM in India carries the onus of cost on themselves and the perceived external stigma around PrEP use [38]. Along with that, considering our research found that discontinuation of PrEP was linked to periods of no sexual activity, more awareness about event-driven PrEP could prove to be further beneficial.

## Limitations

We must acknowledge the limitations of our study. Considering it was purposeful sampling and recruited through online channels, the sample size is not entirely representative of the MSM community-at-large that might be taking PrEP. We also did not interview participants who declined PrEP; hence, the reasons/barriers that led to the decline remain unknown.

## Conclusion

Our study documents real-world experiences of MSM using PrEP in India and provides critical insights that can inform further scale-up, reporting high adherence. Key considerations for further scale-up for PrEP in India would include increasing awareness about PrEP in the community and reducing out-of-pocket expenses to access PrEP Further, the capacities of NGOs providing services to the community should be strengthened and such NGOs should be leveraged for providing PrEP to the community. The interest in LA PrEP should be harnessed and efforts to get LA PrEP in India should be accelerated.

## Acknowledgments

The authors thank UNC-Chapel Hill and NC Tracs staff for their continuous support. We also thank Dr. Siddharth Nandi for supporting data collection in India.

## Author Contributions

**Conceptualization:** Harsh Agarwal, Ivania Núñez, Lauren M. Hill, Karine Dubé, Oluwamuyiwa Pamilerin.

**Formal analysis:** Harsh Agarwal, Ivania Núñez.

**Investigation:** Harsh Agarwal, Lauren M. Hill, Abigail Knoble.

**Methodology:** Harsh Agarwal, Lauren M. Hill, Karine Dubé.

**Project administration:** Harsh Agarwal.

**Resources:** Lauren M. Hill, Karine Dubé, Abigail Knoble, Oluwamuyiwa Pamilerin.

**Software:** Harsh Agarwal, Ivania Núñez.

**Supervision:** Lauren M. Hill.

**Writing – original draft:** Harsh Agarwal.

**Writing – review & editing:** Harsh Agarwal, Lauren M. Hill, Karine Dubé, Abigail Knoble, Oluwamuyiwa Pamilerin.

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
