## [Decision Letter · Decision Letter 0]

26 Jul 2023

PGPH-D-22-01891

Perceptions and Experiences of Daily and Long-Acting Pre-Exposure Prophylaxis (PrEP) among MSM in India

Dear Dr. Agarwal,

Thank you for submitting your manuscript to PLOS Global Public Health. After careful consideration, we feel that it has merit but does not fully meet PLOS Global Public Health’s publication criteria as it currently stands. Therefore, we invite you to submit a revised version of the manuscript that addresses the points raised during the review process.

Please note that we have only been able to secure a single reviewer to assess your manuscript. We are issuing a decision on your manuscript at this point to prevent further delays in the evaluation of your manuscript. Please be aware that the editor who handles your revised manuscript might find it necessary to invite additional reviewers to assess this work once the revised manuscript is submitted. However, we will aim to proceed on the basis of this single review if possible. Please see the reviewer's comments appended below.

The reviewer has identified a number of matters throughout your manuscript that require clarification and/or revision, particularly with regards to separation of the primary results of this study from potential hypotheses. Please ensure you address each of the reviewer's comments when revising your manuscript.

We look forward to receiving your revised manuscript.

Kind regards,

Hugh Cowley

Staff Editor

Journal Requirements:

1. In the ethics statement in the Methods, you have specified that verbal consent was obtained. Please provide additional details regarding how this consent was documented and witnessed, and state whether this was approved by the IRB

2. Please send a completed 'Competing Interests' statement, including any COIs declared by your co-authors. If you have no competing interests to declare, please state "The authors have declared that no competing interests exist". Otherwise please declare all competing interests beginning with twhe statement "I have read the journal's policy and the authors of this manuscript have the following competing interests:"

Additional Editor Comments (if provided):

Reviewers' comments:

Reviewer's Responses to Questions

**Comments to the Author**

1. Does this manuscript meet PLOS Global Public Health’s publication criteria? Is the manuscript technically sound, and do the data support the conclusions? The manuscript must describe methodologically and ethically rigorous research with conclusions that are appropriately drawn based on the data presented.

Reviewer #1: Partly

2. Has the statistical analysis been performed appropriately and rigorously?

Reviewer #1: N/A

3. Have the authors made all data underlying the findings in their manuscript fully available (please refer to the Data Availability Statement at the start of the manuscript PDF file)?

Reviewer #1: Yes

4. Is the manuscript presented in an intelligible fashion and written in standard English?

Reviewer #1: Yes

5. Review Comments to the Author

Reviewer #1: This manuscript contributes to and confirms findings from the published literature base on experiences with oral PrEP in India and globally. The following suggestions for revision primarily aim to clarify and better showcase the key findings from the research.

General

Suggest reviewing for formatting and copyediting, including spacing, punctuation and spelling. There are some inconsistencies and typos throughout.

Introduction

• Suggest adding the year for the HIV sentinel surveillance data in the text body. (“The HIV sentinel surveillance data reported a decrease in HIV prevalence in general population and among MSM at .2% and reported and 3.3% respectively.”) Otherwise it is unclear whether there was a decrease compared to NACO’s data or if they are just from different sources.

• Endnote #4: consider a more recent source, if available.

Methods

• The rest of this sentence seems to be cut off; consider revising: “…our interview guide first explored the participants knowledge of PrEP and initial impressions of PrEP followed by an exploration of their key motivators to initiate PrEP and (including what aspects of PrEP were appealing to them).”

• Suggest condensing the paragraph beginning “Post that…” to improve readability and comprehension. It does not need to present all of the questions asked, but rather provide a sense of the major themes covered in the interview guide.

• Suggest removing or clarifying the question mark in the middle of this sentence: “All interview recordings in English were transcribed verbatim? by professional transcribers.”

Results

• As you’ll see in the comments below, there are areas of the Results that provide potential hypotheses but are not fully supported by the primary data and would be better in the Discussion section with corresponding citations.

• The first couple of sub-sections with qualitative findings could be condensed and tell more of a story to improve flow and comprehension. The latter sections do this well. See specific suggestions below.

Key Motivators to Use PrEP

• In this section, it is unclear which motivators came out most strongly and which were only cited by one or few participants. Suggest clarifying which were raised by more participants so that readers have a sense of the greatest motivators, and perhaps narrow the focus of this section.

• Graduate/post-graduate information was included in the sentence prior to: “Participants mostly resided in urban metropolitan areas of India, were either graduate or post graduate, belonged to general caste, and had an income.” Consider removing here.

• Suggest moving the following to the “Participant Demographics” section: “13 participants were using PrEP at the time of the interview, 4 participants had used PrEP in the past and were no longer using it, and 1 participant reported using PrEP on Demand model. The median duration on PrEP was 10.5 months (minimum being 1 month and maximum being 48 months).”

• Is this sentence speculative or did it come up in the primary research? “Such fear could be possible because of lack of proper knowledge about HIV and the lifesaving antiretroviral treatments that make living with HIV manageable and provide similar quality of life to those who are HIV negative.” Suggest keeping this section to only findings supported by the data collected. If there is supporting evidence from other published literature, this sentence could be moved to the Discussion section.

• The following sentence would read better in the Discussion section with supporting evidence, as it appears to extrapolate based on the quote but is not what the participant stated. “Participant sharing this provides an insight into the existing discrimination that people living with HIV have to face when it comes to seeking sexual or romantic partners, something that could further lead to reduced self-confidence, isolation and adverse mental health outcomes among those who live with HIV.”

• Add a citation to the following sentence and consider moving it to the Discussion section: “This touches upon the lives of many bisexuals and closeted gay individuals who maintain sexual relations with other male partners but prefer to marry a woman, have kids and start a family to meet the social norms and expectations.”

Experiences Accessing PrEP

• Suggest further grouping some of these findings to improve flow and comprehension, focusing on the findings that came out most strongly.

• Add supporting evidence to this sentence if available, or expand to indicate that this has been seen in other settings: “…since PrEP could be equated with promiscuous behavior.” Consider moving to the Discussion section.

• Suggest combining the following sentence with the one prior to it (above the quote, beginning with “Home delivery...”) as they are saying similar things: “Participants appeared to enjoy and appreciate the convenience and option to either get it home delivered or to pick up from the pharmacy.”

• Consider this as an opening paragraph to orient the reader: “Participants’ experiences differed as per the route through which they accessed PrEP. However, the most common route to seek PrEP was through NGOs that offered PrEP at subsidized or no cost.”

Concerns and Dislikes around PrEP

• Consider rephrasing “indulge in risky sexual behaviors” throughout this paragraph as this could imply a value judgement and the field is moving away from risk-based framing.

PrEP Use and dual protection

• Is there a citation/supporting literature you can add here?: “This alludes to the increased use of aphrodisiacs in urban Indian MSM communities…”

Acceptability of Long-Acting PrEP

• Suggest changing “muscular injection” to “intramuscular injection.”

Discussion

• Suggest including the number of participants in some of the studies referenced, particularly those with larger numbers of participants, to indicate how applicable or prevalent certain findings are.

• Add citations to the following sentences if available: “The supreme court of India decriminalized same-sex relationships in 2018, but the battle for equal rights to marry continues in India. These reflections also contradict the usual misconceptions and stereotypes about gay community being promiscuous.”

• Last paragraph on on-demand PrEP: consider adding that your study found that discontinuation of PrEP was linked to periods of sexual activity as another reason to raise awareness for event-driven PrEP use.

Conclusion

• Suggest unpacking and clarifying what this question is implying: “However, at the same time, would the community at large continue to show high adherence if PrEP becomes cheaper and further available?” Is it saying that making PrEP more accessible would reduce adherence, and if so, why is that the takeaway based on what was presented in this study?

6. PLOS authors have the option to publish the peer review history of their article (what does this mean?). If published, this will include your full peer review and any attached files.

**Do you want your identity to be public for this peer review?** For information about this choice, including consent withdrawal, please see our Privacy Policy.

Reviewer #1: No

---

## [Decision Letter · Decision Letter 1]

24 Nov 2023

PGPH-D-22-01891R1

Perceptions and Experiences of Daily and Long-Acting Pre-Exposure Prophylaxis (PrEP) among MSM in India

Dear Dr. Agarwal,

Thank you for submitting your manuscript to PLOS Global Public Health. After careful consideration, we feel that it has merit but does not fully meet PLOS Global Public Health’s publication criteria as it currently stands. Therefore, we invite you to submit a revised version of the manuscript that addresses the points raised during the review process.

We look forward to receiving your revised manuscript.

Kind regards,

Siyan Yi, MD, MHSc, PhD

Academic Editor

Journal Requirements:

Additional Editor Comments (if provided):

Reviewers' comments:

Reviewer's Responses to Questions

**Comments to the Author**

1. If the authors have adequately addressed your comments raised in a previous round of review and you feel that this manuscript is now acceptable for publication, you may indicate that here to bypass the “Comments to the Author” section, enter your conflict of interest statement in the “Confidential to Editor” section, and submit your "Accept" recommendation.

Reviewer #1: (No Response)

2. Does this manuscript meet PLOS Global Public Health’s publication criteria? Is the manuscript technically sound, and do the data support the conclusions? The manuscript must describe methodologically and ethically rigorous research with conclusions that are appropriately drawn based on the data presented.

Reviewer #1: Yes

3. Has the statistical analysis been performed appropriately and rigorously?

Reviewer #1: Yes

4. Have the authors made all data underlying the findings in their manuscript fully available (please refer to the Data Availability Statement at the start of the manuscript PDF file)?

Reviewer #1: Yes

5. Is the manuscript presented in an intelligible fashion and written in standard English?

Reviewer #1: Yes

6. Review Comments to the Author

Reviewer #1: Thank you for the work that was put into this revision. The authors have incorporated most feedback from the previous round of peer review. I have included a few additional, minor comments for consideration.

General

While noting that the revision included a thorough copyedit, I suggest doing an additional review for consistency, as there are still some typos, missing periods, and other formatting errors throughout.

Methods

• Please correct the spelling of “HIPPA” to “HIPAA.”

Results

Participant Demographics

• In Table 1, correct misalignment of participant numbers in the righthand column. In addition, consider changing “Top (Prefer to penetrate)” and “Bottom (Prefer to get penetrated)” to “Top (Prefer insertive anal sex)” and “Bottom (Prefer receptive anal sex),” respectively.

Key Motivators to Use PrEP

• Consider revising “these real conversations” to “these conversations” or “conversations covering these issues.”

Experiences Accessing PrEP

• For “Sources of information,” suggest starting with lines 20-42 (the paragraphs beginning: “Participants identified multiple sources from which they first heard about oral PrEP...”). This would reorganize the section in more sequential order, starting with where participants learned about PrEP followed by their experiences once they tried to access it. To this end, I suggest moving the sentence, “Few participants felt that it was hard to find PrEP in India when they started searching and that PrEP was expensive, which appeared to be one of the barriers to initiating or to stay on PrEP,” and corresponding quote to the end of the Sources of Information section.

• To improve flow, consider moving the sentence “Few participants reported providers to be rude or robotic” (line 36 on p. 10) to appear alongside the corresponding data on p. 11, beginning with the sentence, “One participant felt that their provider was robotic and that the provider was in rush…”

• Suggest combining the following two sentences into one, as they are making similar statements: “Home delivery seemed to be the preferred mode for accessing PrEP, along with pick up from pharmacies. Participants appeared to enjoy and appreciate the convenience and option to either get it home- delivered or to pick- up from the pharmacy.”

Concerns and Dislikes around PrEP

• Suggest changing “initial” to “initiated” in the following sentence: “However, very few reported any side effects once they initial oral PrEP.”

Adherence and Trust with PrEP

• In lines 9-41 on p. 13, trust is described as being fostered before a participant initiates PrEP. Consider moving this section to “Experiences Accessing PrEP” where sources of information and experiences with providers are discussed.

Cost and Affordability

• The following quote is missing attribution to a participant: “So I wouldn't say it's like super cheap also but it's affordable for me and but the concern could be that, for some people out there it could be too expensive. Even at the subsidized rate.”

Community and Peers’ Reactions to PrEP Use

• Lines 22-46 on p. 16 pertain less to the wider community and are more focused on condom negotiation with sexual partners. They could therefore be moved to the “PrEP Use and Dual Protection” section or cut if duplicative.

• The following sentence is speculative; suggest removing or moving to the Discussion section: “This could be because of a lack of access on their part but at the same time having the desire to indulge in unprotected sex.”

7. PLOS authors have the option to publish the peer review history of their article (what does this mean?). If published, this will include your full peer review and any attached files.

**Do you want your identity to be public for this peer review?** For information about this choice, including consent withdrawal, please see our Privacy Policy.

Reviewer #1: No

---

## [Editor Report · Decision Letter 2]

11 Dec 2023

Perceptions and Experiences of Daily and Long-Acting Pre-Exposure Prophylaxis (PrEP) among MSM in India

PGPH-D-22-01891R2

Dear Mr Agarwal,

We are pleased to inform you that your manuscript 'Perceptions and Experiences of Daily and Long-Acting Pre-Exposure Prophylaxis (PrEP) among MSM in India' has been provisionally accepted for publication in PLOS Global Public Health.

Best regards,

Siyan Yi, MD, MHSc, PhD

Academic Editor